# Effective Scheme for Inductive Wireless Power Coil Design Using Scan-and-Zoom Optimization

**Seung-Ha Ryu, Chanh-Tin Truong and Sung-Jin Choi ***

Department of Electrical, Electronic and Computer Engineering, University of Ulsan, 93 Daehak-ro, Nam-gu, Ulsan 44610, Republic of Korea; oscar130@naver.com (S.-H.R.); chanhtin990@gmail.com (C.-T.T.)
* Correspondence: sjchoi@ulsan.ac.kr

**Abstract:** In inductive power transfer (IPT) systems, the coil design is crucial since the power transfer efficiency (PTE) of IPT depends on the coil characteristics such as geometric shape, diameter, wire thickness, etc. The most commonly used technique for the coil is finite element analysis (FEA). Nevertheless, if there are more than two parameters to be designed, FEA design requires a long simulation time since the coil design problem is separated into a series of single-parameter optimization problems. Another issue of conventional FEA is difficulty in interfacing with circuit simulation. To mitigate this issue, a novel co-simulation framework of MATLAB/ANSYS Maxwell is proposed in this paper. In MATLAB, multi-dimensional optimization algorithms like scan-and-zoom are employed to determine geometric parameters to achieve high PTE and minimize the number of FEA executions while Maxwell serves to extract the circuit parameters from the geometric parameters and enhance the accuracy of calculation. The 100 W prototype IPT system is built to verify the proposed coil design scheme in this paper. The performance comparisons with the conventional methods in terms of design accuracy, simulation time, and application flexibility are performed on a pair of designed single-layer circular coils.

**Keywords:** inductive power transfer (IPT); finite element method (FEM); finite element analysis (FEA); scan-and-zoom; optimal coil design

## 1. Introduction

Wireless power transfer technology has recently become widely used in a variety of applications. Inductive power transfer (IPT) in Figure 1 utilizes electro-magnetic coupling that has a low coupling coefficient and thus a high leakage inductance [1–3]. For this reason, it is difficult to achieve a high power transfer efficiency (PTE) when the transmitter ($Tx$) and receiver ($Rx$) coils are directly connected to the circuit and compensation circuits are connected to each of the $Tx$ and $Rx$ coils to make an impedance compensation [4]. Therefore, it is important to design the $Tx$ and $Rx$ coils to maximize the PTE considering the compensation circuit along with the coil geometric parameters.

One of the most popular coil design methods is to directly use the inductance formula. The optimal coil design using the Neumann formula is introduced in reference [5] to calculate the self-mutual inductance of the coil. However, it is too difficult to obtain the general inductance and mutual inductance equation for complex coil shapes such as DD, DDQ, Bi-polar, and Tri-polar [6,7]. In reference [8], the coil is designed based on the ideal circuit parameter values from the Wheeler formula for the single-layer circular coil. However, mutual inductance should be obtained by trial and error. Another equation-based method in reference [9] utilizes the equation obtained by the curve fitting to the measurement data of the single-layer circular coil. In summary, the equation-based methods in [5,8,9] have the following common disadvantages:

- If the coil shape is changed, the inductance equation should be reformulated.

- If the ferrite plate is attached to the coil, the accuracy of the inductance formulas will be reduced.

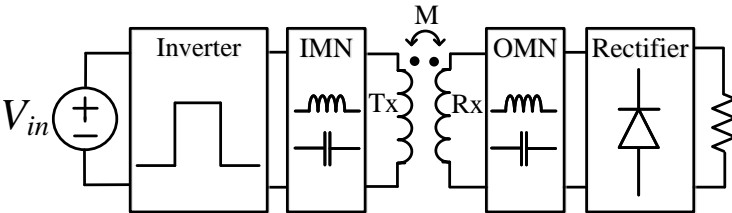

**Figure 1.** Basic IPT configuration with the input-matching network (IMN) and the output-matching network (OMN).

To solve these disadvantages, finite element analysis (FEA) is utilized for the coil design. In the FEA method, the inductance, resistance, and coupling coefficient of the coil can be obtained easily from any coil shape even with the ferrite plate. In reference [10], using the FEA program, the inner and outer radii of the coil are swept to find the high PTE of the coil. The coil is designed via the Pareto front considering the multi-objective function based on the efficiency and the power density [11,12]. However, the FEA method usually requires a long simulation time because the coil or ferrite plate shape parameters should be swept. This disadvantage is also pointed out in reference [13] and a faster coil design method is presented by introducing a lumped-loop model for the FEA simulation instead of conventional brute-force parameter sweeping. However, lumped-loop modeling is difficult to apply to arbitrary shapes and thus lacks generality.

To address the challenge of simultaneously achieving design accuracy, simulation time, and application flexibility in the FEA-based coil design, it is essential to consider the adoption of an optimization algorithm. By integrating an optimization algorithm into the coil design process, the objective is to efficiently mitigate the complexities associated with reducing design simulation speed and increasing accuracy. A crucial aspect of the optimization algorithm's ability is to minimize the number of unnecessary FEA executions, therefore streamlining the coil design process and enhancing its overall power transfer efficiency. In this paper, the scan-and-zoom algorithm, one of the search-based optimization algorithms, is applied to the coil design to achieve a high PTE. The optimization algorithm is implemented in MATLAB and is interfaced with ANSYS Maxwell, one of the most popular FEA programs, to realize the proposed coil design method.

The structure of this article is as follows. In Section 2, a detailed explanation of the conventional FEA-based coil design process is provided. Section 3 presents the coil modeling in the FEA program, the scan-and-zoom algorithm, and the proposed coil design method. The hardware is built to demonstrate the validity of the proposed method in Section 4. Finally, Section 5 provides a conclusion.

## 2. Conventional FEA-Based Coil Design Method in IPT System

As mentioned before, the FEA program tool can analyze various coil shapes. In addition, it is possible to easily obtain inductance, coupling coefficient, or other parameter values when the coil is combined with a ferrite plate or other shielding material. However, the conventional FEA-based method has two major limitations.

One of the limitations is that the conventional method involves too many FEA simulations and thus it takes a lot of time to complete the design. Since the coil design involves multiple design parameters such as the inner and outer radii, or thickness of the wire, the performance should be evaluated for all combinations of the parameters.

Another limitation is that it utilizes a single-turn coil model. In reference [10], the FEA-based coil design method mainly analyzes the single-turn coil model to calculate the self-permeance and mutual permeance of the coil. If the number of turns $N_{Tx}$ and $N_{Rx}$

for the transmitter and the receiver are given, the circuit parameter value can be obtained through (1)–(5).

$$L_{Tx} = N_{Tx}^2 L_{Tx0} \tag{1}$$

$$L_{Rx} = N_{Rx}^2 L_{Rx0} \tag{2}$$

$$R_{Tx} = N_{Rx}^2 R_{Tx0} \tag{3}$$

$$R_{Rx} = N_{Rx}^2 R_{Rx0} \tag{4}$$

$$M = N_{Tx} N_{Rx} M_0 \tag{5}$$

where $L_{Tx0}$ and $L_{Rx0}$ are the single-turn inductance, $R_{Tx0}$ and $R_{Rx0}$ are the single-turn resistances, $M_0$ is the single-turn mutual inductance, and $L_{Tx}$, $L_{Rx}$, $R_{Tx}$, $R_{Rx}$, and $M$ is the circuit parameter considering the number of turns. If the skin effect and proximity effect are not considered, the resistance can be calculated as [14].

In a brute-force sweep method as shown in Figure 2a, every combination in the design parameter space is scanned one by one. However, it requires a lot of time for the FEA calculations. Another way of doing so is to sequentially sweep the parameter as shown in Figure 2b. In this method, one parameter is swept to evaluate the performance and choose the optimal parameter value while other parameters are fixed. Then, the next parameter is swept to repeat this process. However, such a sequential search does not always guarantee the optimal coil design of the coil because a multi-dimensional problem is difficult to tackle with multiple single-dimensional solutions.

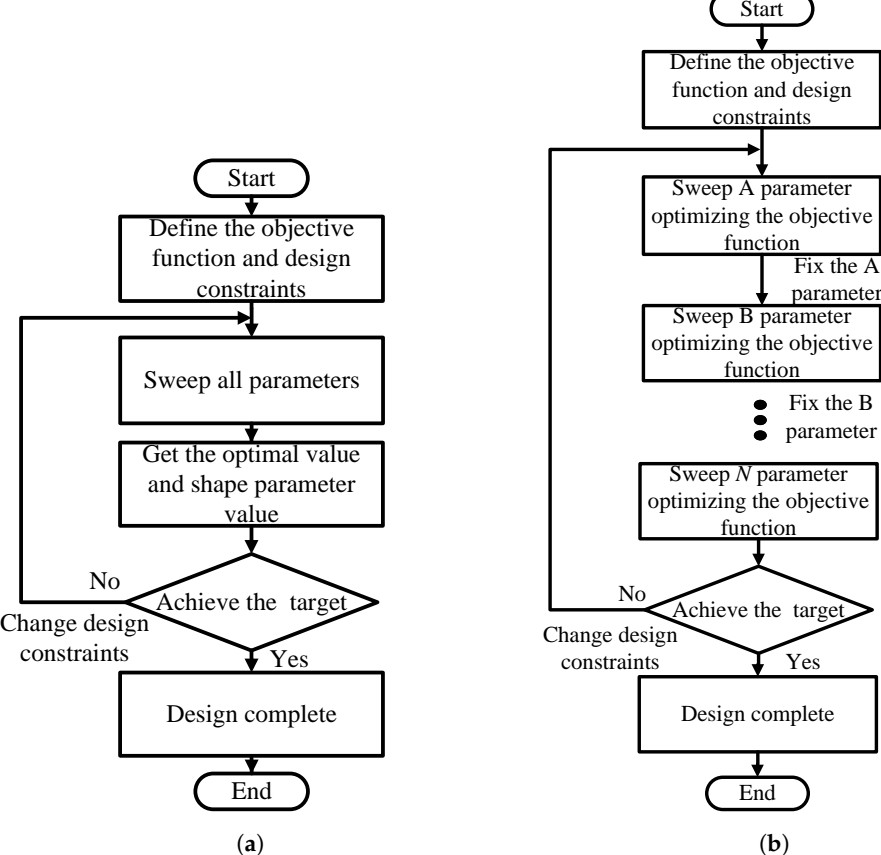

**Figure 2.** The conventional IPT coil design flowchart: (**a**) Brute-force sweep method (**b**) Sequential sweep.

In summary, the conventional single-turn-based FEA is neither accurate nor time-efficient, which motivates a novel coil design method using an optimization algorithm that will be suggested in the next section.

## 3. Proposed Coil Design Method

The FEA-based coil design methods described in the previous section have the disadvantage of taking a long time to obtain an accurate solution. A novel way of mitigating those limitations is proposed in this section.

### 3.1. Effective Coil Modeling Considering the Multiple-Turn Effect

It is well known that a multiple-turn coil has different characteristics from a single-turn coil due to the proximity effect [15]. The difference makes calculation errors, especially for the winding resistance and thus efficiency calculation in the IPT system. To consider the multi-turn effect, there are two different approaches. One is to draw the exact shape of the conductor in the FEA tool, but this increases the number of meshes and thus increases the computation burden. If the number of turns is changing, it requires another model of construction of the coil. Therefore, this kind of approach is not suitable for the coil shape parameter design purpose. The more effective way is to consider the Litz wire effect in the single-conductor model. In this paper, the accuracy of the AC resistance value calculation in the FEA program is greatly increased by adopting the latter way.

### 3.2. Scan-and-Zoom Algorithm

An appropriate optimization algorithm should be chosen to reduce the number of FEA executions in the FEA-based method. However, most gradient-based optimization algorithms require closed-form equations in FEA design problems. For this reason, the scan-and-zoom algorithm, one of the search-based optimizations, is adopted in this paper. The scan-and-zoom algorithm finds the optimal value by narrowing the search range while increasing the resolution and does not require the computation of closed-form equations for the gradient of the cost function. The algorithm is terminated by the following termination conditions: (1) The difference between the best value of the previous step and the best value of the current step is within the tolerance, or (2) the distance between the neighboring search points reaches within acceptable resolution. The scan-and-zoom algorithm will be described below. First, the optimization problem is defined as follows.

$$\begin{aligned} \max / \min f(X) \\ s.t\ X_{lower} \leq X \leq X_{upper} \end{aligned} \tag{6}$$

The center point $X_{ci}$ is chosen and $\Delta X_i$ specifies the search span for determining the search region. The maximum number of search levels $(nL)$ and the number of search points per level $(nP)$ is defined. In each search level, $i$ is stored and the optimal solution $f_{i,opt}(X_{i,opt})$ and the location $X_{i,opt}$ of the optimal solution in that level are stored after scanning the number of search points. Finally, the termination conditions are checked if $\Delta f_i \leq \varepsilon_1$ or $\Delta R_1 \leq \varepsilon_2$, where

$$\Delta f_i = \left| f_{i,opt}\left(X_{i,opt}\right) - f_{i-1,opt}\left(X_{i-1,opt}\right) \right| \tag{7}$$

$$\Delta R_i = \frac{\left| X_{i,upper} - X_{i,lower} \right|}{nP - 1}. \tag{8}$$

$$X_{i,upper} = X_{ci} + \Delta X_i \tag{9}$$

$$X_{i,lower} = X_{ci} - \Delta X_i \tag{10}$$

$$\Delta X_i = \frac{X_{upper} - X_{lower}}{2} \tag{11}$$

If the condition is satisfied, then the algorithm is terminated. Otherwise, the center point will be chosen to be equal to the optimal solution point and the search region is zoomed by reducing $\Delta X$ by $\Delta X_i$ half. The algorithm flow chart is summarized as shown in Figure 3.

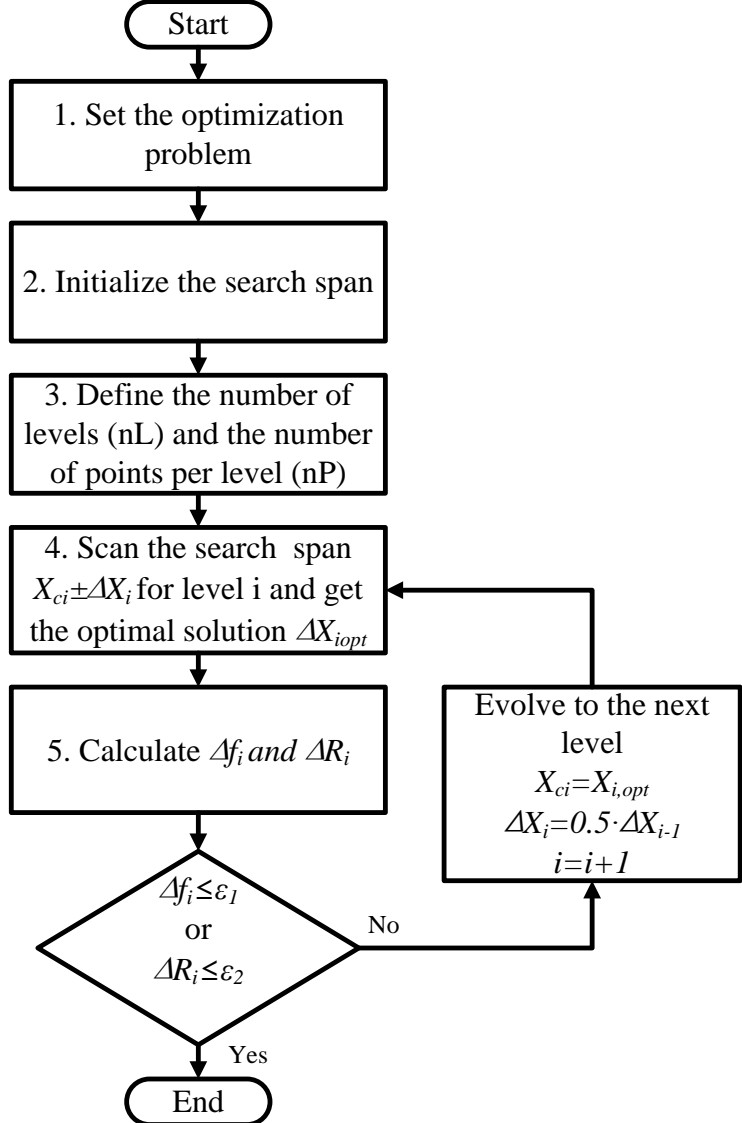

**Figure 3.** Scan and zoom algorithm.

### 3.3. Proposed Coil Design Method

In this section, the proposed coil design method using the scan-and-zoom algorithm is explained. The proposed coil design flowchart is shown in Figure 4 and the detailed design process is shown as follows:

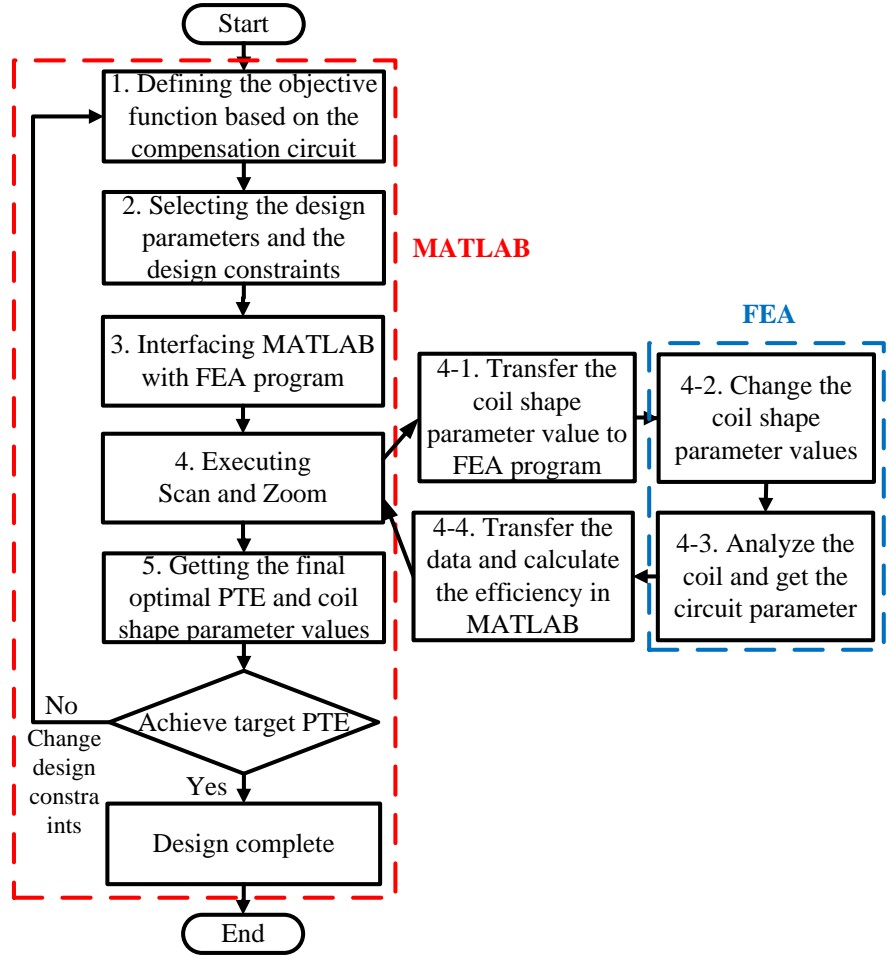

**Figure 4.** The proposed IPT coil design flowchart.

Step (1) Defining the objective function: At first, the target inductive power transfer and the compensation circuit are decided. For example, a series–series compensation circuit can have higher PTE than other compensation circuits and but another compensation network such as series–parallel, parallel–series, or parallel–parallel can also be considered [16,17]. When the compensation network is determined, the objective function can be defined as the system performance index. Typically, PTE is the objective function that can be maximized. The PTE formula is usually obtained by solving the circuit equations and is determined by the compensation network [18].

Step (2) Selecting the design parameters and the design constraints: Design parameters can be determined after the coil shape is selected. Design constraints are the parameters that the designer limits for the system design such as the coil radius, air gap, wire diameter, etc. Typically, circular coils can achieve high PTE and high coupling coefficient [19]. In this shape, the outer and inner radii can be designed as parameters and the design constraints are specified by the overall coil size.

Step (3) Interfacing MATLAB with ANSYS Maxwell: To calculate the objective function, the coil shape parameter values and the circuit parameter values are exchanged between MATLAB and ANSYS Maxwell, as shown in Figure 5. This process can be automated by interfacing interface MATLAB with ANSYS Maxwell using the script. MATLAB should compute PTE and run the optimization algorithm and ANSYS Maxwell extracts the circuit parameter values such as the inductance, resistance, and coupling coefficient by analyzing the coil in design. To automatically implement this process, the script function of ANSYS Maxwell is internally called MATLAB script.

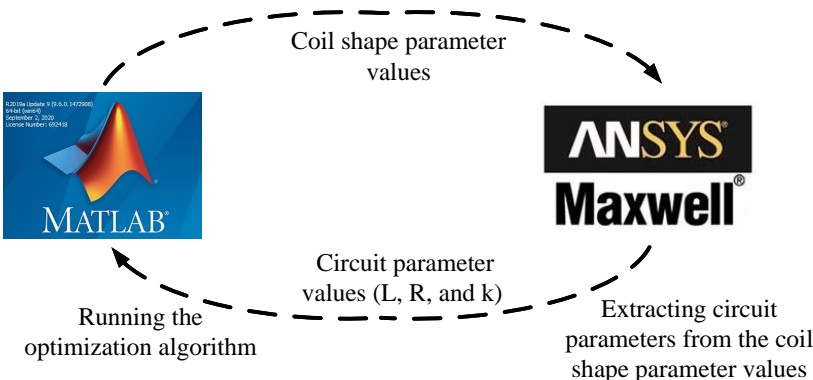

**Figure 5.** Concept of the co-simulated design process.

Step (4) Executing the scan-and-zoom algorithm: The scan-and-zoom algorithm is implemented and executed through MATLAB. However, since the circuit parameter values corresponding to each coil shape parameter value are required to calculate the PTE, MATLAB sends the information of the coil shape parameter values during the optimization. The circuit parameter values are then extracted by ANSYS Maxwell and sent back to MATLAB. At this time, the PTE is calculated through the circuit equations, and it is repeatedly executed until the termination condition in Section 3.2 is met.

Step (5) Obtaining the PTE and coil shape parameter value results and checking the target PTE.

After the algorithm is terminated, the maximum PTE and the corresponding coil shape parameter values are obtained. If the maximum PTE obtained from the algorithm does not reach the target PTE defined in Step 2, the initial coil design constraints are released to satisfy the target *PTE*. If the target PTE is met, the design is completed and *Tx* and *Rx* coils are configured with the corresponding coil shape parameter values to build the IPT system.

Indeed, the proposed method offers another distinct advantage by utilizing the circuit simulation tool in MATLAB (MATLAB Simulink) as shown in Figure 6. This advantage allows for the consideration of semiconductor losses, which cannot be directly accounted for in the optimization function of ANSYS Maxwell. By incorporating the circuit simulation tool, we can more accurately evaluate the impact of semiconductor losses on the overall system performance, providing a comprehensive analysis that enhances the reliability and effectiveness of the optimization process. This capability further reinforces the superiority of the proposed method in capturing real-world performance aspects that might be overlooked using ANSYS Maxwell alone.

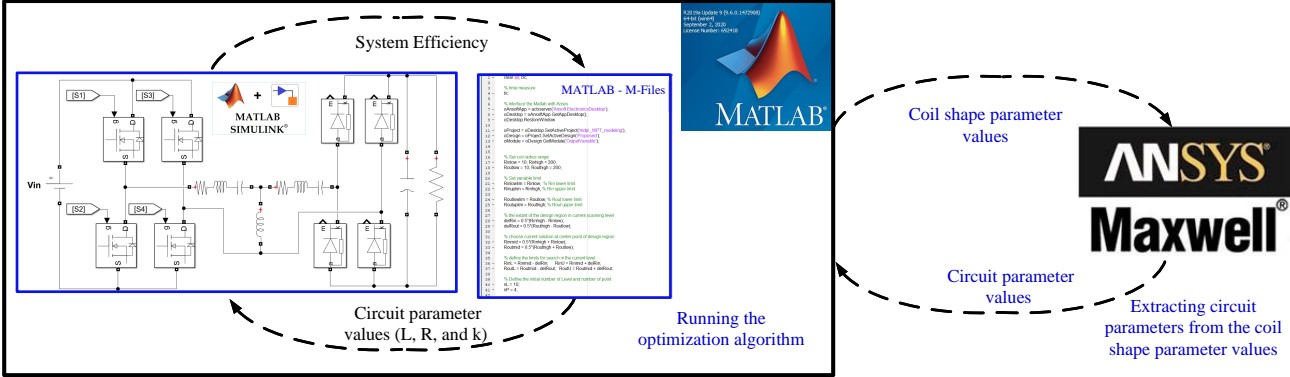

**Figure 6.** Concept of the co-simulated design process with circuit simulation integration.

Thermal design is another critical consideration in the development of the IPT system, particularly for high-power and compact setups [20–22]. Addressing temperature rise is a crucial aspect that can be incorporated into the optimization procedure. In the future,

with slight modifications, this design procedure can be easily applied to take into account thermal considerations, enabling the development of more efficient and robust thermal designs for the IPT system.

## 4. Performance Comparison and Verification

This section shows a design example and hardware verification for a pair of single-layer circular coils with the radius constraint of 10 mm $\leq R_{in} \leq R_{out} \leq$ 200 mm. The performance of the proposed coil design method is compared with the conventional FEA-based coil design method.

### 4.1. Design Examples and Performance Comparisons

For a design example, a single-layer circular coil in Figure 7 is selected and a case where the coil size and circuit parameter values are symmetrical is considered. In this case, the coil shape parameter only considers two coil shape parameters: the outer coil radius $R_{out}$ and the inner coil radius $R_{in}$, therefore the problem is simplified. Among the compensation circuits in IPT, a series–series compensation circuit is selected. For the symmetrical series–series compensation circuits as shown in Figure 8, the PTE equation at the resonance point can be obtained as (12), which is used for the calculation of the objective function [18].

$$\eta = \frac{k^2 Q^2 r_d}{(1 + r_d)^2 + k^2 Q^2 (1 + r_d)} \tag{12}$$

where $k$ is the coupling coefficient, $Q$ is the quality factor of the coil and $r_d$ is the ratio between of equivalent load resistor and coil parasitic resistance. Coil modeling in ANSYS Maxwell is generated as a single-turn coil as shown in Figure 9, and the coil size and ferrite core size are adjusted within the design constraints. In addition, when using a single-layer circular coil, wire diameter $d_w$ is assumed to be equal to the coil thickness $t_w$. The coil is made of about 3.4 mm diameter Litz wire with 500 strands of 0.10 mm thickness. Therefore, $t_w$ is around 3.4 mm. The Litz wire information is updated in ANSYS Maxwell by the number of conductors in the coil cross-section.

$$N = \frac{R_{out} - R_{in}}{t_w} \tag{13}$$

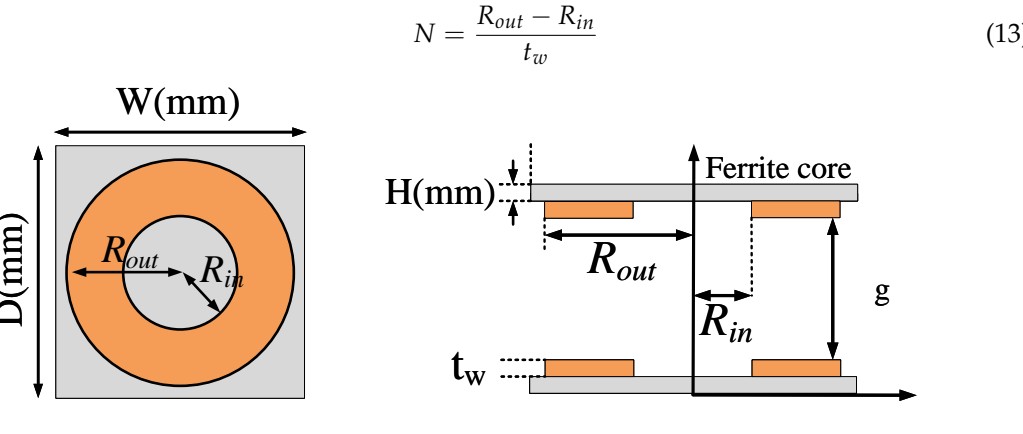

(a)                                                                                            (b)

**Figure 7.** Single-layer circular coil (**a**) Top view (**b**) Side view.

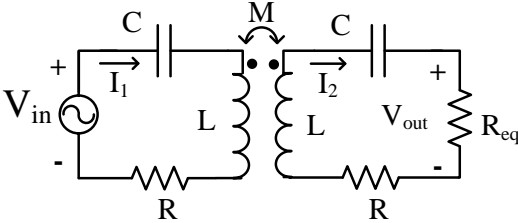

**Figure 8.** Simplified circuit diagram for the symmetric series–series compensation circuit.

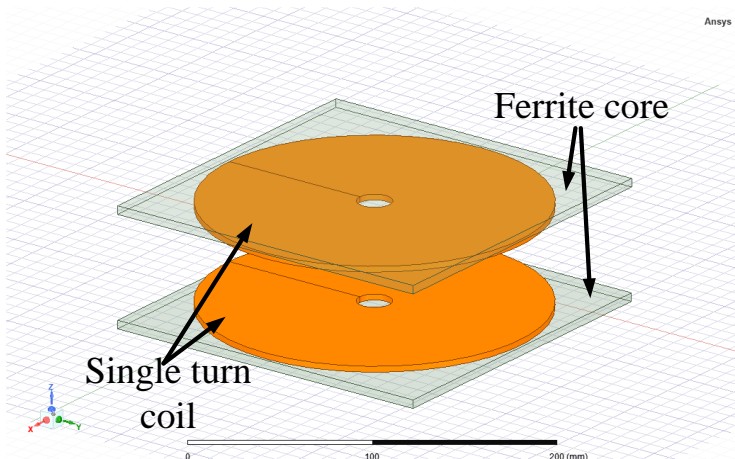

**Figure 9.** Single-turn coil modeling in ANSYS Maxwell for verification.

For the IPT system design requirements, the output power $P_{out}$ is set to 100 W, the output voltage $V_{out}$ is 50 V, and the operating frequency $f_0$ is 100 kHz. Through $P_{out}$ and $V_{out}$, the load resistance $R_L$ is set to 25 Ω. For the target performance, the target *PTE* $\eta_t$ is set to 95% or higher. This information is summarized in Table 1. The design constraints are set in Table 2 and the optimization problem is defined as an efficiency maximizing problem as shown in (14).

$$
\begin{aligned}
maximize \quad & \eta(R_{out}, R_{in}) \\
subject\ to \quad & 10\ \text{mm} \le R_{in} < R_{out} \le 200\ \text{mm}
\end{aligned}
\tag{14}
$$

To examine the performance of the algorithm, the conventional FEA-based coil design method and the proposed coil design method are applied and compared together.

**Table 1.** IPT system design requirements.

| Symbol | Parameters | Value | Unit |
|--------|------------|-------|------|
| $P_{out}$ | Output power | 100 | W |
| $f_0$ | Operating frequency | 100 | kHz |
| $V_{out}$ | Output voltage | 50 | V |
| $\eta_t$ | Target efficiency | ≥95 | % |

**Table 2.** Design constraints for IPT.

| Symbol | Parameters | Value |
|--------|------------|-------|
| $R_{out}$ and $R_{in}$ | Outer and inner radius | $10\ \text{mm} \le R_{in} < R_{out} \le 200\ \text{mm}$ |
| $g$ | Coil separation/distance | 95 mm |
| $d_w$ | Wire diameter | 3.4 mm |
| $t_w$ | Coil thickness | 3.4 mm |
| $W \times D \times H$ | Ferrite dimension | 400 mm × 400 mm × 4 mm |

### 4.2. Conventional FEA-Based Coil Design Method

The conventional FEA-based coil design method in reference [10] is adopted to take this problem as shown in Figure 10. First, $R_{in}$ is fixed at 10 mm which is a minimum coil radius constraint, and $R_{in}$ is swept by the step size of 1 mm. The resulting search trajectory is shown in Figure 11a, and the optimal coil radii, maximum PTE, and the number of FEA executions are shown in Table 3.

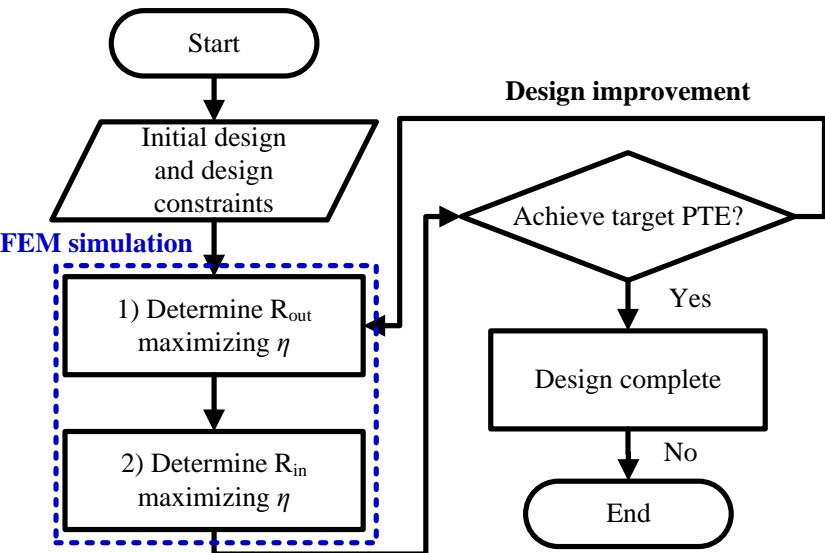

**Figure 10.** Simplified flow chart of the conventional coil design method.

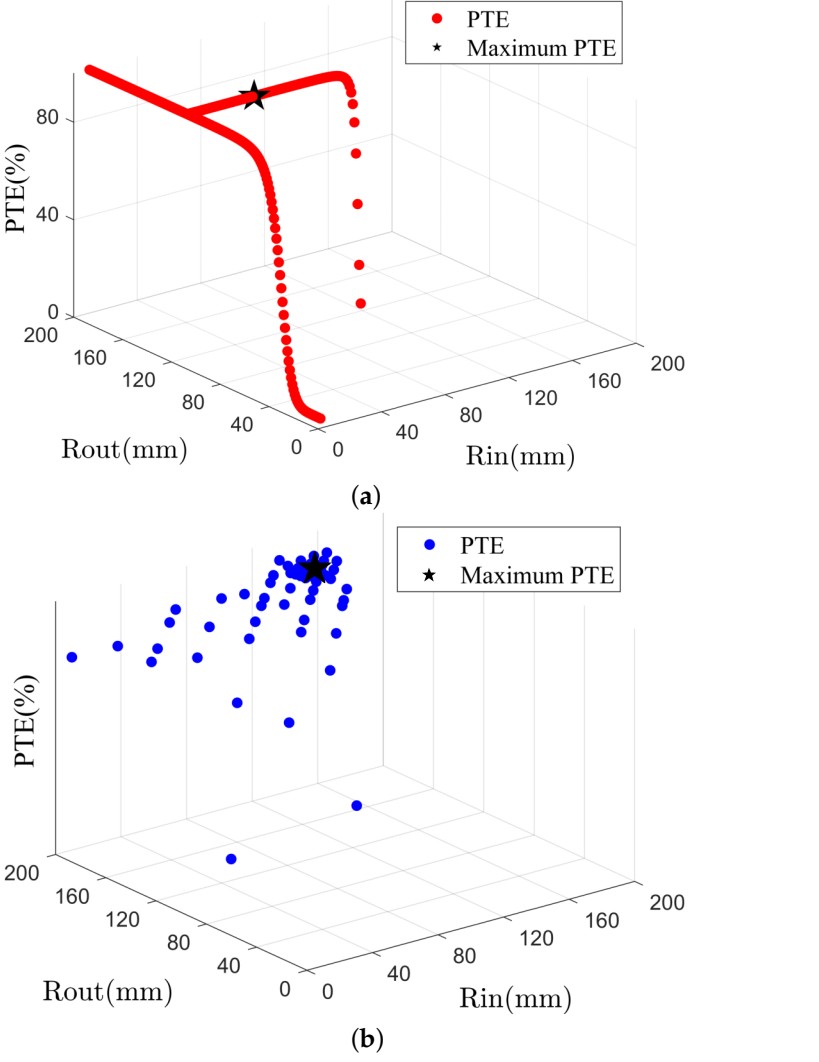

**Figure 11.** The search trajectory (**a**) Conventional FEA-based coil design (**b**) Proposed coil design method.

**Table 3.** The optimization results of coil design methods.

| Parameters | Conventional [10] | ANSYS Maxwell (Pattern Search) | Proposed Co-Simulation |
|---|---|---|---|
| Outer radius $R_{out}$ | 120 mm | 146.25 mm | 174.271 mm |
| Inner radius $R_{in}$ | 52 mm | 102.5 mm | 138.646 mm |
| PTE (coil) | 99.88%/99.25% (without/with Litz wire option) | 99.406% (with Litz wire option) | 99.448% (with Litz wire option) |
| # of FEA executions | 299 | 56 | 64 |
| Simulation time * | 14,950 s | 2800 s | 3200 s |
| PTE (overall) | 99.76%/99.13% | N/A | 99.3353% |

* In both the conventional and proposed methods, simulation time is recorded using the MATLAB tic-toc function, while the ANSYS simulation time is determined through actual execution. All evaluations were conducted on an Intel Core i7-8600K processor.

Another approach to implementing the optimization is to use the optimization function in ANSYS Maxwell. Pattern search is selected for comparison with the proposed method. This is because the scan-and-zoom algorithm and the Pattern Search algorithm share several similarities such as:

- Non-Gradient: Both algorithms are derivative-free optimization techniques, meaning they do not require gradient information of the objective function. This makes them suitable for optimizing functions that are non-smooth or non-differentiable.
- Simplicity: Both algorithms are relatively easy to implement compared to more complex optimization techniques. This simplicity makes them accessible choices for solving optimization problems.
- Direct Search: Both algorithms perform a direct exploration of the search space. They systematically evaluate the objective function at different points to locate the optimal or near-optimal solution.
- Global Exploration: Both methods can explore a wide range of the search space. This ability to avoid becoming trapped in local optima makes them suitable for problems with complex landscapes.
- Constraints Handling: Both algorithms can be adapted to handle constrained optimization problems where the solution needs to satisfy certain constraints.

The optimization results of coil design methods are shown in Table 3. The optimization function in ANSYS Maxwell is embedded in the same software with the finite element analysis (FEA), which makes the speed of data transfer faster than the proposed method. Therefore, the execution time of the optimization function in ANSYS Maxwell is slightly faster compared to the proposed method. However, the power transfer efficiency (PTE) of ANSYS Maxwell optimization is lower compared to the proposed method. This is because the proposed method is implemented in the M-file of MATLAB, which provides high flexibility to optimize the code for the WPT system. Moreover, the proposed method can cooperate with the circuit simulation in MATLAB Simulink. Therefore, all losses from the circuit simulation such as switching losses and conduction losses of MOSFET and diode are fully considered in the proposed optimization method, resulting in a more accurate optimization process.

*4.3. Proposed Coil Design Method*

The input parameters of the optimization algorithm are $R_{in}$ and $R_{out}$ as shown in Table 2 and the PTE is the output parameter of the optimization algorithm. To apply the proposed coil design method, nL is set to 10, and $nP$ is set to 4. The scan-and-zoom algorithm is terminated at level 5 by the termination criterion of $\varepsilon_1 = 10^{-4}$ % and $\varepsilon_2 = 5\,\text{mm}$ as shown in Figure 12. The results are summarized in Table 3, and the search trajectory is shown in Figure 11b.

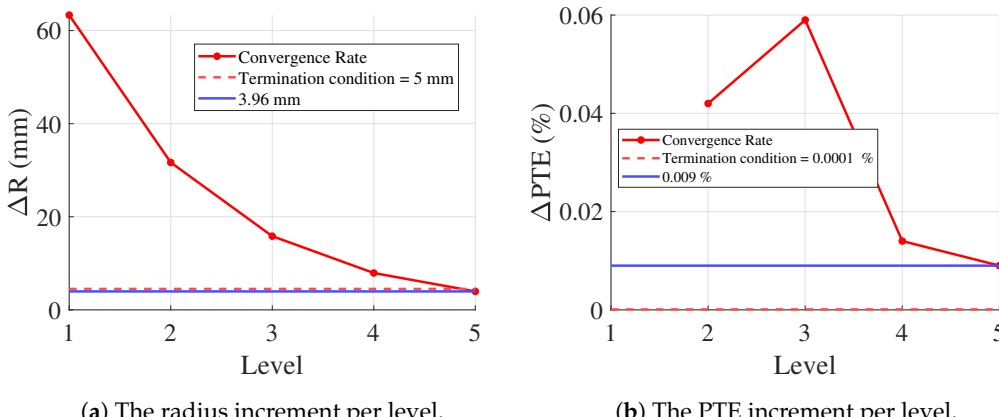

(**a**) The radius increment per level.

(**b**) The PTE increment per level.

**Figure 12.** Convergence rate of (**a**) the radius increment per level (**b**) the PTE increment per level.

The number of FEA executions of the proposed coil design method is much lower than the conventional coil design method as shown in Table 3. The PTE of the conventional coil design method without the Litz wire option is higher than the proposed coil design method because the internal resistance of the coil is underestimated by ignoring the skin and proximity effects. The PTE calculated by the Litz wire option in the conventional coil design method is 99.25%. Because of these results, the conventional coil design method not only takes a long time to extract the coil shape parameter values but also reaches non-optimal solutions.

Semiconductor losses are a crucial factor that significantly affects power transfer efficiency. However, most existing optimization methods do not take these losses into account. In the proposed optimization method, it becomes effortless to consider semiconductor losses by utilizing circuit simulation in MATLAB. This allows for the inclusion of all losses from the circuit simulation, leading to a more comprehensive and accurate optimization process. To further substantiate the superiority of the proposed method, we implemented MATLAB Simulink circuit schematics in Figure 13, which calculates the overall PTE by the time simulation. The detailed non-ideal MOSFET and diode device characteristics are shown in Table 4. By incorporating the circuit simulation tool, we can more accurately evaluate the impact of semiconductor losses on the overall system performance, providing a comprehensive analysis that enhances the reliability and effectiveness of the optimization process. This capability further reinforces the superiority of the proposed method in capturing real-world performance aspects that might be overlooked using ANSYS Maxwell alone.

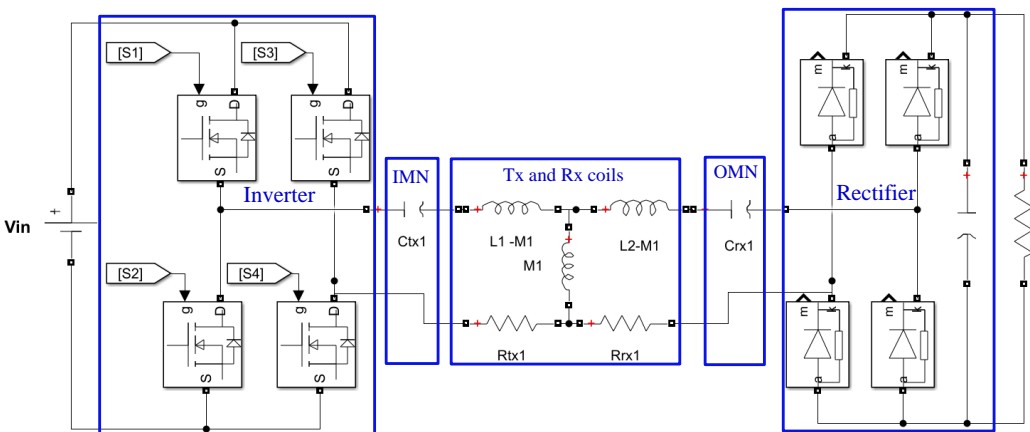

**Figure 13.** MATLAB Simulink circuit schematics.

**Table 4.** MATLAB Simulink circuit parameters.

| Components | Description |
|---|---|
| MOSFET | Drain-Source on-state resistance $R_{DS(on)}$= 0.4 $\Omega$, internal diode forward voltage $V_f$ = 4.4 V |
| Diode | On resistance $R_{on}$ = 0.1 $\Omega$, forward voltage $V_f$ = 1.6 V |

## 5. Hardware Verification

To further validate the previous comparison results, a 100 W prototype IPT system is constructed. The IPT configuration is depicted in Figure 14, and the experimental setup is shown in Figure 15. LCR meter (Agilent 4263B) is used to measure the inductance, resistance, and capacitance, while the power analyzer (Yokogawa WT1804E) is used to measure the PTE. Wireless power transfer (WPT) systems are commonly operated at resonant points to optimize their performance and efficiency. The resonant operation allows for efficient power transfer between the transmitter and receiver coils by minimizing losses and ensuring maximum power transfer at specific frequencies. Therefore, in both the proposed and conventional methods, the switching frequency is set equal to the resonant frequency. Both the coils, designed by either the conventional method or the proposed method, are built for a fair comparison.

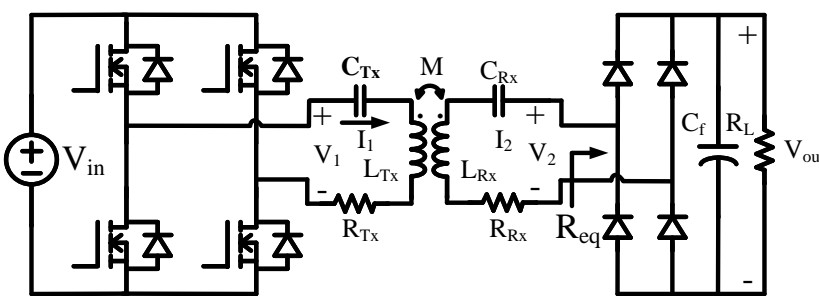

**Figure 14.** IPT configuration schematic for hardware.

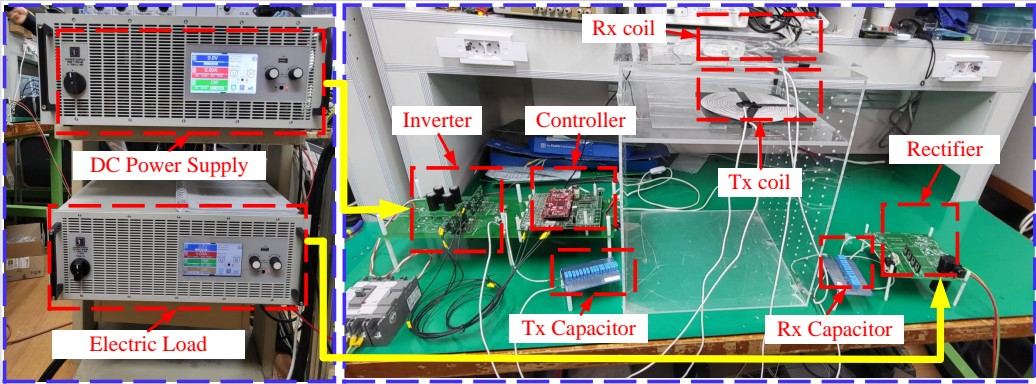

**Figure 15.** Experimental setup.

### 5.1. Evaluation of Conventional FEA-Based Coil Design

According to conventional method in Section 4.2, $R_{out}$, $R_{in}$, and N are set equal to 120 mm, 52 mm, and 20, respectively. Table 5 provides a summary of the FEA results with and without the Litz write option, and the measurement value of coils as shown in Figure 16. Table 6 shows the calculated and measured capacitance based on the measured inductance results of the coil. The resistance in the FEA results without the Litz write option is highly inaccurate. In ANSYS Maxwell, the Litz write option creates a stranded effect at the coil cross-section, but it ignores the skin effect. However, the accuracy is further improved.

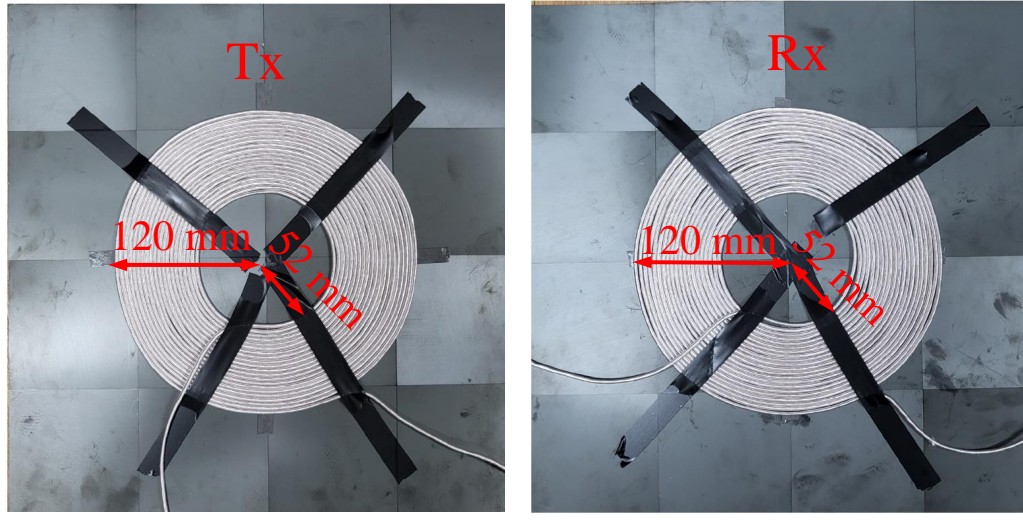

(**a**) Transmitter coil.                    (**b**) Receiver coil.

**Figure 16.** Transmitter and receiver coils using coil radius of the conventional coil design method.

**Table 5.** FEA and measured results using coil radius ($R_{out}$ = 120 mm and $R_{in}$ = 52 mm) of the conventional coil design method.

| Parameters | FEA Results Using with Litz Wire Option | FEA Results without Litz Wire Option (Error) | Hardware Test (Error) |
|---|---|---|---|
| $L_{Tx}$ | 158.61 μH | 158.61 μH (0%) | 159.4 μH (+0.5%) |
| $L_{Rx}$ | 158.61 μH | 158.61 μH (0%) | 156.56 μH (−1.29%) |
| $R_{Tx}$ | 101.53 mΩ | 16.22 mΩ (−84.02%) | 147.42 mΩ (+45.2%) |
| $R_{Rx}$ | 101.53 mΩ | 16.22 mΩ (−84.02%) | 143.42 mΩ (+41.51%) |
| $k$ | 0.294 | 0.294 (0%) | 0.263 (−10.54%) |

**Table 6.** Calculated and measured capacitance results of proposed and conventional coil design method.

| Symbol | Parameters | Calculated Results | Measured Results (Error) |
|---|---|---|---|
| $C_{Tx}$ | Transmitter capacitance | 15.89 nF | 15.92 nF (−0.19%) |
| $C_{Rx}$ | Receiver capacitance | 16.18 nF | 16.07 nF (−0.68%) |

*5.2. Evaluation of Proposed Coil Design*

According to the proposed design in Section 4.3, $R_{out}$, $R_{in}$, and N values in the proposed coil design approach are set equal to 173 mm, 139 mm, and 10, respectively, as shown in Figure 17. Tables 6 and 7 summarize the FEA and measured results, as well as the capacitance results. Table 8 summarizes the experimental results. The $P_{out}$, $V_{out}$, and $f_o$ values are determined following the IPT system design specifications. The efficiency is measured when the system is operated at resonance frequency $f_o$ where the current and voltage phases are in phase as shown in Figure 18. The proposed coil design method shows higher efficiency than the conventional coil design method by 0.401%. Furthermore, the proposed method has fewer FEA executions than the conventional method.

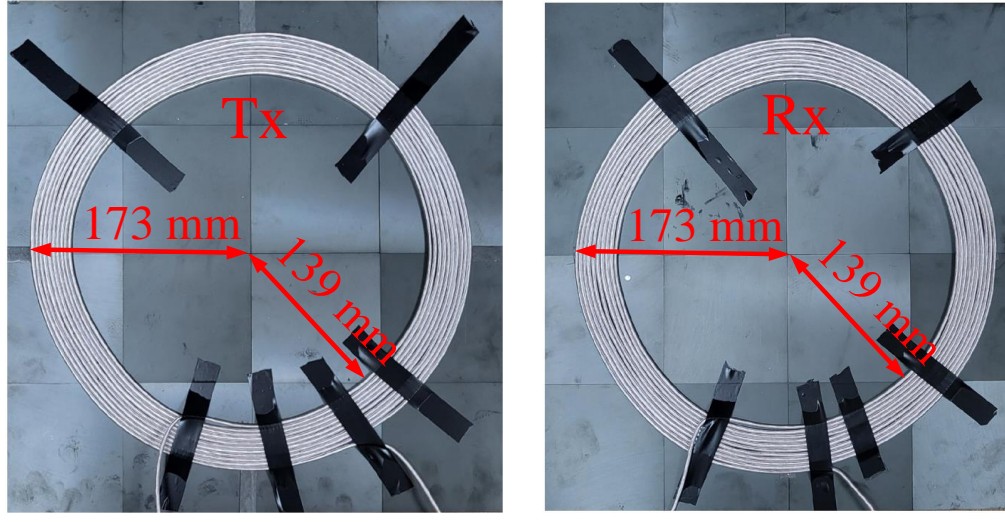

(**a**) Transmitter coil.          (**b**) Receiver coil.

**Figure 17.** Transmitter and receiver coils using coil radius of the proposed coil design method.

**Table 7.** FEA and measured results using coil radius ($R_{out}$ = 173 mm and $R_{in}$ = 139 mm) of the proposed coil design method.

| Parameters | FEA Results | Measured Results (Error) |
|:---:|:---:|:---:|
| $L_{Tx}$ | 117.28 μH | 117.3 μH ($+0.02\%$) |
| $L_{Rx}$ | 117.28 μH | 115.32 μH ($-1.67\%$) |
| $R_{Tx}$ | 70.19 mΩ | 92.83 mΩ ($+32.26\%$) |
| $R_{Rx}$ | 70.19 mΩ | 89.86 mΩ ($+28.0251\%$) |
| $k$ | 0.349 | 0.29 ($-5.73\%$) |

**Table 8.** The experimental results.

| Symbol | Parameters | Conventional | Proposed |
|:---:|:---:|:---:|:---:|
| $P_{out}$ | Output power | 100 W | 100 W |
| $f_0$ | Operating frequency | 100.6 kHz | 101.3 kHz |
| $V_{out}$ | Output voltage | 49.5 V | 49.4 V |
| $R_L$ | Load resistance | 24.89 Ω | 24.93 Ω |
| $\eta_t$ | Power transfer efficiency | 97.734% | 98.135% |

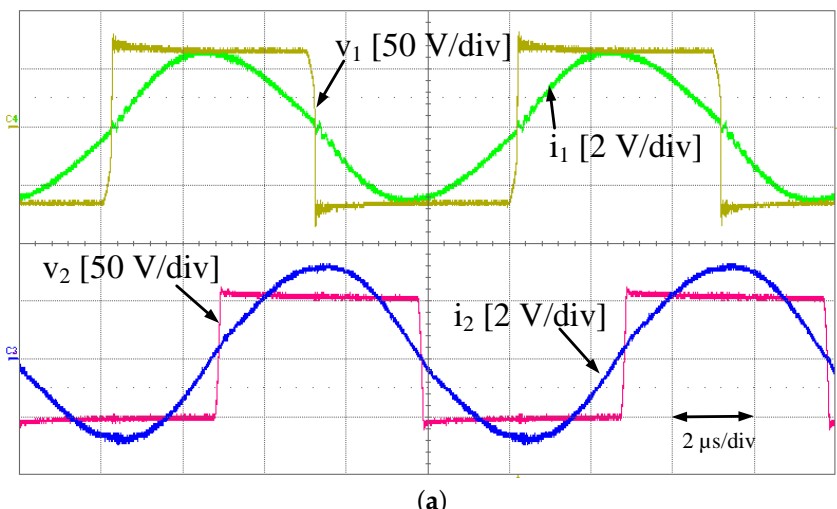

(**a**)

**Figure 18.** *Cont.*

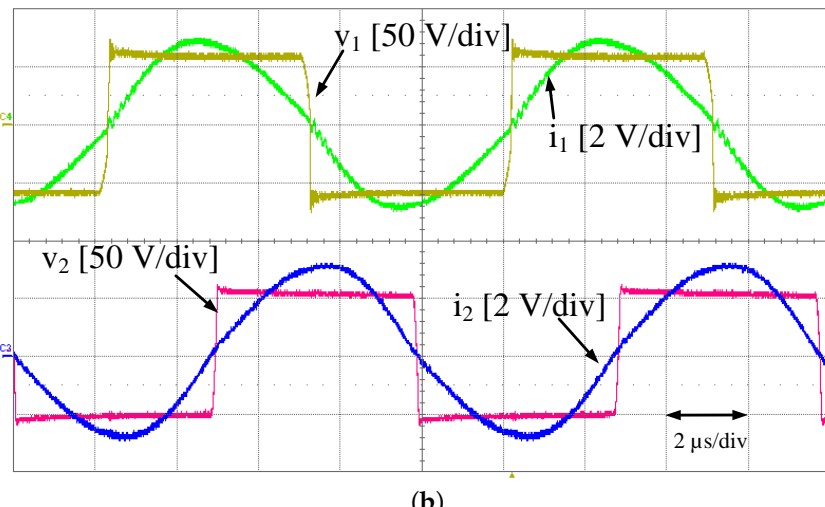

**Figure 18.** Experiment waveforms of (**a**) conventional method (**b**) proposed method.

## 6. Conclusions

This paper introduces an effective coil design method for the IPT system, which is implemented using a co-simulation framework of MATLAB/Maxwell. The proposed method employs the scan-and-zoom algorithm to optimize the PTE of IPT coils, while simultaneously reducing design simulation speed and increasing accuracy. The experiment verification results for a 100 W IPT system demonstrate the superior performance of the proposed method, as it significantly outperforms the conventional FEA-based coil design approach in terms of PTE. This comprehensive analysis and validation provide strong evidence of the practicality and effectiveness of the proposed coil design method, making it an asset for IPT system design applications. Additionally, the proposed method offers a distinct advantage by utilizing the circuit simulation tool such as MATLAB Simulink. By incorporating the circuit simulation tool, we can more accurately evaluate the impact of semiconductor losses on the overall system performance, providing application flexibility in the design process.

**Author Contributions:** Conceptualization, S.-J.C.; Software, S.-H.R.; Formal analysis, S.-H.R.; Writing—original draft, S.-H.R.; Writing—review & editing, C.-T.T. and S.-J.C.; Supervision, S.-J.C. All authors have read and agreed to the published version of the manuscript.

**Funding:** This work was supported by Regional Innovation Strategy (RIS) through the National Research Foundation of Korea (NRF) funded by the Ministry of Education (MOE) (2021RIS-003).

**Institutional Review Board Statement:** Not applicable.

**Informed Consent Statement:** Not applicable.

**Data Availability Statement:** Not applicable.

**Conflicts of Interest:** The authors declare no conflict of interest.

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
