# Peer review of "Effective Scheme for Inductive Wireless Power Coil Design Using Scan-and-Zoom Optimization"

_applsci, doi:10.3390/app13169299_

Round 1

Reviewer 1 Report

This article introduces an innovative algorithm designed to optimize coils in wireless power transfer systems, resulting in a significantly reduced need for FEA simulations. The authors assert the superior efficiency of their proposed method compared to traditional optimization approaches. This is exemplified through a detailed case study, where the application of their algorithm reduced the requisite number of simulation runs by over four times.

Major Points:

The comparative analysis provided in Table 3 appears to be inherently flawed. The primary concept of reference [9] does not revolve around an optimization algorithm, hence it seems inappropriate to compare it with the proposed optimization algorithm in this paper. A comparison with other extant optimization algorithms would yield a more equitable and meaningful evaluation. Various optimization methods are incorporated in ANSYS Maxwell and HFSS 2023 R2's optimization window. Of particular relevance would be the search-based method, which provides a fitting benchmark for evaluating the authors' algorithm. The following list includes the optimization algorithms available in ANSYS Electronics:

1-Screening (Search based)
2-Multi-Objective Genetic Algorithm,
3-Nonliner Programming by Quadratic Lagrangian (Gradient) 
4-Mixed-Integer Sequential Quadratic Programming (Gradient and Discrete) 
5-Adaptive Multiple Objective 
6-Adaptive Single Objective (Gradient)

Minor Points:

  • Further evidence is required to substantiate the superiority of this optimization algorithm over alternative methods employed in wireless power transfer systems.
  • The authors should delineate the input and output parameters of the optimization algorithm, along with its convergence rate.

Reviewer 2 Report

1.Valuable work for WPT researchers and engineers, since the FEA design requires large time and computing resources. 

2. There is an error for the arrow direction in Fig. 4. since no information is transferred from FEA software to Matlab.  

3. Comparision of the computing time requirement of the proposed method and conventional method should be provided.  

Reviewer 3 Report

The reviewed article is interesting and it could be published after major revision. In the revised version of this article the Authors should take into account following remarks:

1. The aim of the article should be clearly formulated in the section Introduction.

2. This section should not start by a figure. Firstly, some text should be given.

3. In the literature review the papers https://doi.org/10.3390/en15197236 and https://doi.org/10.3390/en16031391 should be included.

4. Some information about the thermal problems in designing of air transformers should be added in Introduction.

5. In Hardware verification an influence of frequency on the parameters of the tested device should be considered and presented.

6. The section Conclusion should be extended. The advantages of the proposed design method should be clearly pointed.

Round 2

Reviewer 1 Report

You have addressed the previous suggestions well. This will make your comparison more valid and convincing. Table 3 shows that the built-in optimization method in ANSYS Maxwell performs better than proposed algorithm. However, both your algorithm and the HFSS method outperform the conventional method. This is acceptable because you have proposed an alternative way of optimizing the design at the cost increased complexity.

There are also some minor changes that need to be addressed. 
1- complete table 3 and specify the type of optimization method used by ANSYS Maxwell. Did you use the search-based method, which is the most similar to yours?

2-Please explain table 3 in more detail and compare your proposed methods with the ANSYS Maxwell optimization algorithm. How do they differ in terms of performance, efficiency, and accuracy?

Reviewer 3 Report

The Authors properly addressed all my remarks. The paper could be accepted in its current form.

Author Response

Thank you for your positive feedback regarding the revisions made to address your comments.